# Review of Machining Equipment Reliability Analysis Methods based on Condition Monitoring Technology

**Wei Dai** [1] , **Jiahuan Sun** [1], **Yongjiao Chi** [2], **Zhiyuan Lu** [3] , **Dong Xu** [4],*** and **Nan Jiang** [5]

1    School of Reliability and Systems Engineering, Beihang University, Beijing 100191, China
2    System Safety Dept, Beijing Branch, CASCO SIGNAL LTD, Beijing 100160, China
3    School of Mechanical Engineering and Automation, Beihang University, Beijing 100191, China
4    School of Automation Science and Electrical Engineering, Beihang University, Beijing 100191, China
5    Beijing Institute of Control Engineering, Beijing 100195, China
*    Correspondence: xd@buaa.edu.cn; Tel.: +86-132-6023-1660

**Abstract:** The condition of mechanical equipment during machining is closely related to the accuracy and roughness of the workpiece. In an intelligent sensing environment, a large amount of multi-source data reflecting status information are generated during processing, and a number of studies have been conducted for machining equipment reliability analysis. In this paper, the reliability analysis method of machining equipment based on condition monitoring technology is taken as the main line. And an up-to-date comprehensive survey of multi-source information during the cutting process, failure physical analysis for signal selection and reliability assessment based on condition information will be provided. Finally, the future challenges and trends will also be presented. It is a feasible and valuable research direction to evaluate the reliability of machining equipment for product quality characteristics.

**Keywords:** machining equipment reliability; condition monitoring technology; multi-source information

## 1. Introduction

A regular analysis of machining equipment reliability not only reduces production maintenance costs, but also ensures the machining quality of the workpiece. The traditional reliability analysis methods use a large number of similar device failure time information to obtain their overall characteristics, which is based on the experimental data of a lot of samples, and the result is only an average property under the current conditions. The reliability analysis method based on a degradation model is a breakthrough in the development of reliability engineering technology. But there are still some weaknesses, such as the degraded data in practical is difficult and costly to obtain. In recent studies, state monitoring methods have been used to obtain dynamic signals, which reflect the health state of the equipment and can be used for reliability analysis without disturbing the normal operation of the equipment.

For machining equipment, high productivity and required product quality ultimately need to be achieved. Therefore, the reliability analysis of machining equipment should consider the mission reliability based on equipment status identification, process design analysis and product quality. All of the above methods focus on the operational reliability of the equipment, while ignoring the characteristics of the mechanical machining equipment that is different from the general mechanical equipment. In recent years, some scholars have studied the task reliability of machining equipment by applied the multi-source data of the processing process.

In this paper, the reliability analysis is summarized from three aspects: Multi-source information during cutting process, failure physical analysis for signal selection and reliability assessment based on condition information. The technical method of condition monitoring is combed according to the framework of reliability analysis. This paper focuses on how to extract relevant features from the cutting process information and how to use the multi-source data to perform real-time reliability assessments of machining equipment.

Based on the above content, a literature search was carried out. The databases referenced in this paper mainly include "Web of Science", "Google scholar", and "China National Knowledge Infrastructure". The number of keywords to be searched in Web of Science and the number of documents in past five years are shown in Table 1. It can be seen that "Condition Monitoring Technology" and "Quality Control" have been the hot topics of current research, and related researches of other keywords also show an upward trend.

**Table 1.** Keyword search summary results.

| Key Words | 2014 | 2015 | 2016 | 2017 | 2018 |
|---|---|---|---|---|---|
| Machining Equipment and Reliability and Quality | 10 | 15 | 15 | 16 | 16 |
| Multi-Source Data Fusion | 17 | 14 | 18 | 41 | 49 |
| Cutting process and Failure mechanism | 22 | 32 | 47 | 49 | 55 |
| Equipment and Reliability Assessment | 65 | 65 | 70 | 58 | 89 |
| Condition Monitoring Technology | 531 | 587 | 678 | 748 | 887 |
| Cutting Process and Quality Control | 116 | 149 | 150 | 172 | 180 |

According to the search results, the rest of this article is organized as follows: Section 2 reviews the multi-source information during cutting process. The failure physical analysis for signal selection is described in Section 3. In Section 4, the detail of reliability assessment based on condition information are presented. Finally, the conclusions of this work and outlook on future challenges are provided in Section 5.

## 2. Multi-Source Information during Cutting Process

In the intelligent sensing environment, the cutting process of the machining equipment will generate a large amount of data, which contains both static data and dynamic data. According to the data source, it is mainly divided into the following three categories: cutting process information of machining equipment, quality characteristics of processed products, process flow and parameter information.

### 2.1. Cutting Process Information of Machining Equipment

The cutting process information of the machining equipment is dominated by high frequency sensor data. The cutting process is a complex physical process with many types of signals that can be extracted. The current research mainly focuses on cutting force signals, vibration signals, machine power signals, sound signals and acoustic emission signals. They are used to monitor equipment condition and life in a variety of machining formats such as turning, milling, and drilling.

The cutting force signal is generally measured by resistance strain type force gauges and piezoelectric crystal type force gauges. Studies have shown that the change in cutting force is the physical phenomenon most closely related to the tool failure condition. In many studies, the cutting force is decomposed into three directions of X, Y, and Z according to the direction of the spindle feed, which are tangential force, feed force, and radial force. However, the measurement cost of the cutting force signal is high. Additionally, the measurement also has an influence on the cutting process. By collecting the three-direction cutting force signal and applying the Six Sigma principle and factoring the data, Chen Hongtao et al. [1] have proposed an effective method for predicting cutting force in the absence of empirical formulas and testing conditions. This method provides a basis for optimizing

cutting conditions. Oraby S. E. et al. [2] have used nonlinear regression analysis techniques to model the effects of various cutting force components on tool wear and tool life. Cutting force signals are costly to measure and have an impact on the cutting process. At the same time, the cutting force signal is sensitive to processing conditions, processing material characteristics, tool parameters, etc. It is difficult to extract the characteristic information related to the processing state.

The vibration signal is generally collected by a piezoelectric acceleration sensor. Vibration is a low-frequency oscillation caused by periodic changes in the cutting component, resulting in changes in the vibration amplitude and vibration frequency of the machining system [3]. Studies have shown that the vibration signal during the cutting process contains a wealth of information on the wear condition and it is easy to monitor. The vibration signal sensor has a simple structure and low cost, so it is also the main signal in the detection of other mechanical devices. The vibration signal is easily affected by the noise of the processing environment, and is often doped with environmental noise; so filtering is often required for research [4]. Julie Z. Zhang et al. [5] have studied the relationship between the vibration of the spindle and the feed axis and tool wear by using an optical system. This method successfully monitored the change in tool wear condition, but the monitoring accuracy was limited. The vibration signal is easily affected by the noise of the processing environment and the installation position. Filtering is often required for research.

The power and current signals can be measured directly from the machine motor. When the tool wears, the cutting force will increase, and the load on the machine will increase. It can be reflected in the motor that the power and current will increase. Therefore, the measurement of power and current signals avoids the influence of the sensor on the machining process. However, the power and current signals are less sensitive to tool wear. Renede Jesus et al. [6] have established a model between motor current and tool breakage to predict tool breakage. Wang Junping [7] has used stochastic fuzzy neural network to establish the corresponding relationship between tool wear and motor current. This model can identify the tool wear condition in a certain precision range.

The Acoustic Emission (AE) phenomenon was discovered by German scientist Kaiser in 1955. At present, AE technology has been rapidly developed in engineering applications and is recognized as a new monitoring technology with great potential. The Acoustic emission signals, which are high-frequency oscillations generated by the release of strain energy in the form of elastic stress waves when the material is damaged, can be detected by piezoelectric sensors [8]. In the machining process, many acoustic emission signals are generated, which are from the tool wear, the deformation of the workpiece, and the friction of the chip. The information of the tool wear is included in the acoustic emission signal, as shown in Figure 1 [9].

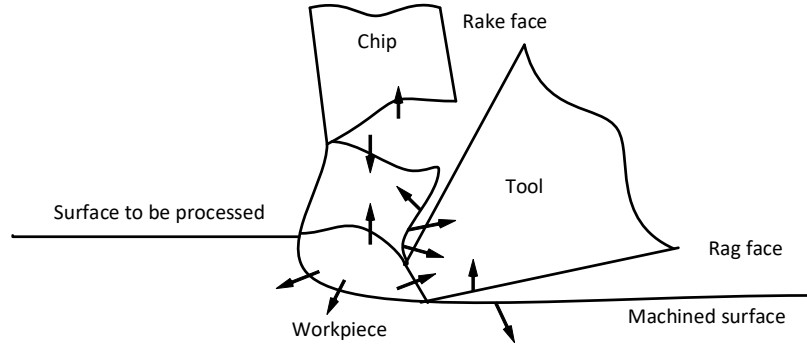

**Figure 1.** Sources of Acoustic Emission (AE) in machining.

As a key component of machining equipment, researchers have done a lot of research on cutting tools. At present, tool wear monitoring technology based on acoustic emission technology has been vigorously promoted, with the advancement of signal processing technology and the rapid development of computer technology. AE technology is recognized as a promising new monitoring technology. Compared with other monitoring methods, the acoustic emission signal has high sensitivity.

The signal frequency (100 kHz~1 Mhz) is higher than the noise frequency, so it avoids the low frequency band (with serious noise) during processing. However, the cost of the sensor is high. Additionally, a high-frequency signal acquisition device is also required [10]. Many researchers have proposed monitoring programs under various processing scenarios. Liu T et al. [11] have used AE to perform tool wear detection during the cutting process. It is considered that the power spectrum of the AE signal increases with the wear of the tool within 350 kHz, and the sum of AE counts is closely related to the tool wear. The results of Xiaozhi Chen and others have shown that under normal wear and tear, AE mainly comes from the first, second and third deformation zones and is a typical continuous signal. But when the tool is damaged, the AE signal is a discontinuous signal [12].

The following studies use the AE signal for wear monitoring of drills, turning tools and milling cutters respectively. The AE signal can be used to meet the requirements of tool condition monitoring in many aspects. Tansell has proposed procedures to detect tool breakage and to estimate tool condition (wear) by using AE. The proposed procedure filters the AE signals with a narrow band-width, band-pass filter and obtains the upper envelope of the harmonic signal by using analog hardware [13]. Xie Jianfeng et al. have studied the milling process of 45 steel, applying wavelet transform to multi-resolution decomposition of acoustic emission signals, and extracting the signal energy of each frequency band as the feature quantity [14]. GómezM P et al. have studied a drilling process with different degrees of wear in the drill bit to find relationships between acoustic emission (AE) and torque measured during the drilling process, and also with the degree of wear of the tool [15]. KJemielniak et al. have studied the roughing process of Inconel, and have used wavelet packet transform to extract characteristic values such as skewness, kurtosis and energy of the AE signal and the wavelet component coefficient of the cutting force signal. The correlation coefficient method was used to optimize the kurtosis and energy, and then the neural network algorithm established a tool wear model based on cutting force and acoustic emission signals [16]. Zhang Dongliang et al. have studied the milling process of aluminum alloy 6061, analyzing the acoustic emission signal by chaotic time series analysis method, extracting the embedding dimension and Lyapunov coefficient of the signal as feature quantities, and establishing a tool wear monitoring model based on vector machine [17]. Hu Jianglin et al. have studied TC4 titanium alloy drilling and the research showed that AE signal ringing number could clearly indicate the tool wear rate [18]. Neslušan et al. have studied the turning of 100Cr6-62 materials, and have used two kinds of AE sensors to separately acquire high-frequency and low-frequency AE signals, and have used the relationship between the two signals as the feature quantity to determine tool wear [19]. Nie Peng et al. have studied the turning process of GH4169, and used statistical methods and wavelet analysis to extract the root mean square of AE signal amplitude and the energy of 10–150 KHz band as the signal feature quantity, and established a tool wear monitoring model based on a wavelet neural network. [20]. Zhou Yumeng has studied the aluminum alloy turning process, extracted the energy of the AE signal 7.8–1.25 KHz band, and established a tool wear monitoring model based on a BP neural network [21]. Maia et al. have studied the turning process of AISI 4046 steel. The research showed that the average density of the power spectrum of the AE signal which is closely contact with wear and tear, is high at the initial value of the tool life. As the tool enters the mid-life value, the level of the end of the life is gradually increased [22].

In addition, other sensing signals during processing can be acquired. Using the current or power of the machine tool as the monitoring parameter of the processing state has the advantages of simple measurement, convenient extraction and low cost [23–25]. Wireless sensor methods have also been widely used in health, environment, home and agriculture sectors [26–29].

## 2.2. Quality Characteristics Information of Processed Products

The quality characteristic information of processed products is mainly based on a class of quality characteristic data with a slower change frequency. This type of data mainly includes processing dimensions, tolerances, ambient temperature and other information that need to be inspected in the

process. There are two main ways to monitor product quality information: Quality control technology based on statistical process control and data-driven product quality monitoring technology.

Statistical Process Control (SPC) is an important research content in product quality control and design. It can reduce the scrap rate caused by fluctuations in various factors in the process. The actual production process usually uses statistical analysis methods to analyze and monitor the measured data of key processes at the site. Therefore, it is possible to grasp the abnormal fluctuation of product quality and take timely measures to control it. The effect of the control chart is largely determined by the recognition of out-of-control conditions from a pattern recognition perspective. Pattern recognition is an important issue in SPC. In the field of process quality control, many researchers use eight control chart modes to represent different processing states. For example, normal mode (NOR), hierarchical mode (STA), system mode (SYS), periodic mode (CYC), upper step mode (US), lower step (DS), trend up (UT), and trend direction Next (DT) [30–36]. In the application of control charts for the processing quality of parts, a lot of exploration has been done. Much analysis and research have been done on the causes and physical meanings of various abnormal control chart modes [37].

Data-driven product quality monitoring methods can generally be divided into direct methods and indirect methods. In the direct method, direct detection of surface quality can be achieved by laser technology, ultrasonic technology, and the fringe field capacitive method [38–40]. However, direct monitoring methods are very sensitive to the test environment. They require special testing equipment. And these devices are very difficult to install and use. For indirect detection methods, people often use some sensing signals related to the manufacturing process, including vibration, cutting force, noise, acoustic emission, temperature, spindle torque, spindle voltage and power, to indirectly monitor the surface quality of the product.

## 2.3. Process Flow and Parameter Information

The processing flow and parameter information are mainly based on a type of data in which the values in the process nodes remain unchanged. This type of data mainly includes product information, equipment basic information, personnel information, process flow and process parameter information. This information remains unchanged in a process node. And it is also the attribution information of the quasi-static information and dynamic information.

The three types of cutting process information are not isolated. They are related to each other. Additionally, the data flow model of the whole cutting process can be constructed. The model relevance is mainly reflected in the following two aspects.

On the one hand, the external sensor information in the cutting process can reflect the quality characteristic information. People usually use some sensing signals related to the manufacturing process, including vibration, cutting force, noise, acoustic emission, temperature, spindle torque, spindle voltage and power to indirectly monitor the product. Although indirect methods are not as accurate as direct methods, they are very convenient to use during actual processing. In this regard, researchers have made a lot of research on the surface quality monitoring of products by using process-sensing signals [41–46].

On the other hand, quality characteristics such as surface roughness are directly or indirectly affected by process parameters and process flows. For example, in roughing, the larger cutting depth $a_p$ is generally selected first, followed by the larger feed rate $f_c$, and finally the appropriate cutting speed is determined according to the tool durability requirements, thus ensuring a high metal cutting rate. In the finishing process, the machining accuracy and surface quality requirements of the workpiece should be determined first. In addition, Szymon Wojciechowski proposes a method for the reduction of forces and the improvement of efficiency during finish ball end milling of hardened 55NiCrMoV6 steel. The primary objective of this work concentrates on the optimal selection of milling parameters, which enables the simultaneous minimisation of cutting force values and increased process efficiency [47,48]

### 3. Failure Physical Analysis for Signal Selection

The cutting process is a complex physical process with many types of signals that can be extracted. The selection and processing of cutting process information is an important part of equipment reliability analysis. There are a lot of differences between these signals, such as the acquisition method, sensitivity to the cutting process and processing method. It is important to select the appropriate signal and method to improve the accuracy of the cutting condition monitoring. Therefore, it is crucial to select the signal that is most sensitive to the form of machining equipment failure by combining the signal selection method with the machining equipment failure mechanism.

Failure physical analysis is the study of the relationship between the various failure phenomena of the machining equipment (failure mode) and the incentives that cause the failure (stress, including environmental stress and time stress). The failure physical analysis for signal selection is to study the internal mechanism of the machining equipment in the event of failure, and find out the influence of the failure mode on various signals during the cutting process. Thus, the occurrence of machining equipment failure can be monitored and determined by selecting an appropriate signal. During the research, the physical signal is converted into an electrical signal by installing a sensor in the processing system, and is transmitted to the computer as a digital signal through an amplifying device and a collecting device, as shown in Figure 2.

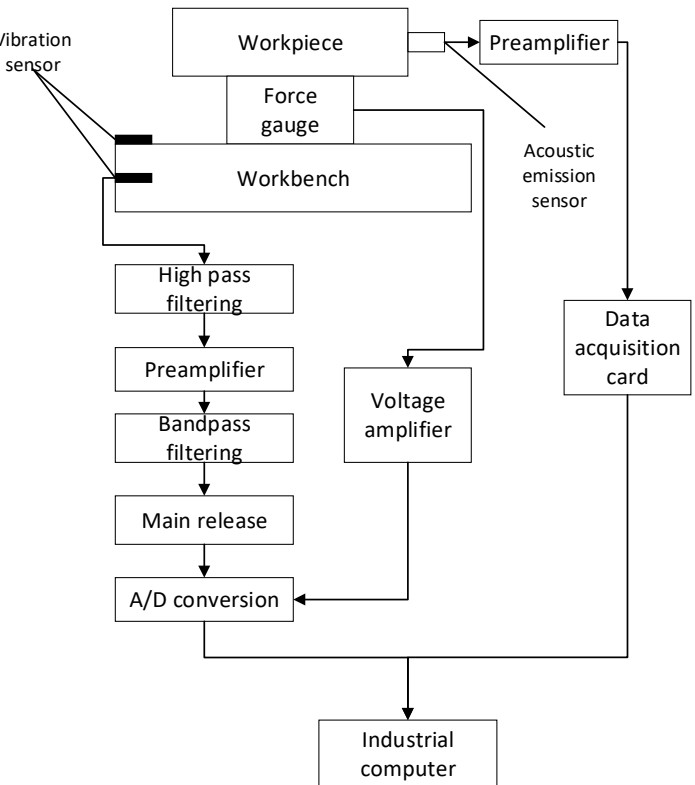

**Figure 2.** Signal acquisition principle.

### 3.1. The Relevance Analysis between Failure Mechanism and Output Signal

Due to the external load, the machining equipment will gradually be damaged during operation. The damaged machining equipment will have a certain influence on the output signal during the cutting process. The relationship between external load, output signal, and machining equipment damage is shown in the Figure 3. In the figure, $Z(t)$ indicates the external load acting on the machining equipment over time, $U(t)$ indicates the degree of damage of the machining equipment over time, and $X(t)$ indicates the change of the output signal of the cutting process with time [49].

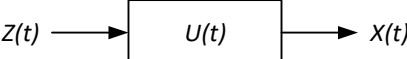

**Figure 3.** Relationship between external load, output signal, and damage.

$Z(t)$, $U(t)$ and $X(t)$ are both quantities that change over time. And their relationship can be expressed as Equations (1) and (2).

$$U(t) = \phi(Z(t)) \tag{1}$$

$$X(t) = y(U(t)) \tag{2}$$

It can be seen from the equation that the relationship between the output signal represented by $X(t)$ and the degree of damage represented by $U(t)$ depends on the mapping affected by the machining equipment structure and the failure mechanism. In addition, the physical properties of the machining equipment material itself are closely related to its damage. The damage is the characterisation of the microscopic physical process of the machining equipment material. And the process signal is a macroscopic reflection of the machining equipment state change process.

For the common defects in the cutting process, the relationship between the failure mechanism and the process signal is summarised. Based on the summary results, the process signal that is most sensitive to a certain failure mode can be selected.

### 3.1.1. Flutter

Flutter is an unstable self-excited vibration that occurs in almost all machining processes. During the cutting process, if the system is subjected to an instantaneous accidental disturbance, the relative vibration of the tool and the workpiece will occur. The amplitude of the vibration will gradually decay due to the presence of the system damping, but the vibration will leave a ring of vibration on the existing machined surface. When the workpiece is rotated one revolution, the tool will cut on the surface with the vibrating surface. As a result, the cutting thickness will be too large or too small, which will result in dynamic cutting force. If the various conditions during the cutting process further promotes the vibration, it will develop into flutter. Strong vibrations also generate a lot of noise, irritating the operators [50].

Because flutter is a strong vibration phenomenon, researchers use the vibration sensor to extract the characteristic quantity from the vibration signal for the monitoring of cutting flutter. In the literature [51], the vibration signal is used to analyze the milling condition. The results show that the vibration signal is more reliable and highly used in related research. In addition, flutter will cause changes in cutting force and generate a lot of noise. Therefore, in addition to the vibration signal, some scholars also use the cutting force and the sound signal. For example, the literature [52] selects the sound signal as the monitoring signal of the milling flutter and establishes the milling flutter monitoring model. The sound signal is more convenient to collect than the cutting force signal and it does not affect the actual machining.

### 3.1.2. Breakage

Tool breakage typically includes crack formation, expansion, instability, or impact fracture processes. For the monitoring method of tool breakage, many scholars use cutting force [53], machine motor current [54], power [25], vibration [55] and acoustic emission [56] monitoring signals. Among these sensing signals, Acoustic Emission (AE) signals have received extensive attention due to their high sensitivity, anti-interference and easy installation.

Two characteristics of the AE signal can be used to determine whether a crack is formed, as shown in Figure 4. The first is the stepwise change of the AE signal count rate when the tool state is gradually worn to severe wear. The second is the sudden AE signal caused by the fracture of the tool material, which has a strong energy. Its amplitude is generally high.

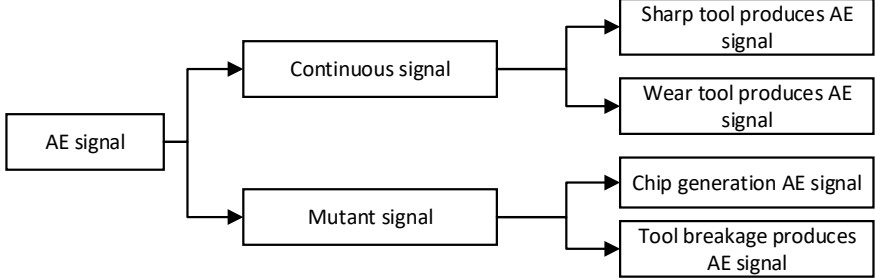

**Figure 4.** Type of AE signal generated during tool damage.

### 3.1.3. Tool Wear

Tool wear is one of the most typical phenomena in the machining process. Numerous studies have shown that there are mechanical, thermal and chemical effects in the tool wear process, as well as friction, adhesion and diffusion. It is the result of the combined action of one or more forms of wear. The main cause of tool wear is mechanical heat and chemical wear [57]. Mechanical wear is caused by the scoring of hard spots in the workpiece material. Thermal and chemical wear is caused by the bond and the diffusion.

When the tool is severely worn, the amplitude and the energy of the AE signal generated by the friction between the tool and the workpiece increases due to the increased contact area between the tool flank and the workpiece. And the frequency components will be diversified. At the same time, due to the complexity of the tool wear form, tool wear can also lead to the increase of cutting forces, cutting temperatures and vibration.

Therefore, scholars usually combine a variety of signals, such as cutting force signals, vibration signals, acoustic emission signals, power or current signals to study the wear state of the tool [58].

### 3.2. Application of Multi-Sensor Information Fusion

Through the study of the mechanism of machining equipment failure, it can be concluded that the damage of the machining equipment is usually the result of a combination of various factors. It is a complex physical phenomenon caused by high temperature, high pressure and huge intermittent impact. Therefore, machining equipment damage is usually accompanied by a variety of signal changes. The application of multi-sensor information fusion is mainly to make an accurate judgment and prediction of the state and real-time reliability of the machining equipment through the comprehensive processing and analysis of various signal data. Moreover, according to the above analysis, various sensors may have certain defects. Therefore, it is usually required to be used simultaneously with other sensors. The signals are mutually verified, so that the processing results are more reliable.

In the open technical literature, multi-sensor information fusion first appeared in the late 1970s. In 1988, the US Department of Defense listed fusion technology as one of the key technologies developed in the 1990s. In 1998, the International Information Fusion Society was established, which promoted the development of source-aware information fusion technology [59]. According to the three abstract levels of multi-sensor information, the level of information fusion can be divided into three levels: Data set fusion, feature set fusion and decision-level fusion. In the fusion algorithm, weighted average method, maximum likelihood estimation, cluster analysis, support vector machine, neural network method, genetic algorithm, Kalman filtering method, Bayesian estimation method, fuzzy set theory, DS evidence theory and other algorithms are formed. In terms of application, at the 23rd Japan International Machine Tool Show in 2006, an intelligent spindle was exhibited by Yamazaki Mazak Company of Japan. The spindle is equipped with various sensors such as temperature, vibration, displacement and distance to protect and warn the spindle. The development goal is to automatically optimize the parameters such as the spindle speed in the future according to the processing conditions [60].

At present, multi-source sensing information fusion technology has been widely used in military and civilian fields, including fault diagnosis, manufacturing process detection and control. Therefore, researchers have also continuously explored the state of tooling in the fusion of multiple signals.

B. Bahr et al. have used vibration and visual sensors to detect and identify the wear condition of tools [60]; Matsushima et al. [61] have proposed the combination of the various sensor signals or at least the sensor information of multiple sensors to evaluate the tool machining state and tool failure; Sohyung Cho of the University of Miami in the United States studied multi-sensor based tool condition monitoring for multi-coated multi-layer carbide end mills for 4340 steel, using multiple sensors for force, vibration, acoustic emission and spindle power sensors. The experimental results show that the TCM design based on feature level fusion can significantly improve the accuracy of tool state classification. Chryssolouris G et al. have applied multi-source sensing information by neural network, group clustering and least squares regression method, and compare the performance of the monitoring system. The experimental results show that the tool monitoring system with multi-source sensor is applied. The accuracy is improved over a single sensor.

Liang Jiancheng [62] et al. have identified the research object as the milling process, and applied the sensor to separately acquire the signal of three directions of force and acceleration, as well as the acoustic emission signals, extracted their feature vectors and used parameter features such as feed rate, Spindle speed and depth of cut, etc., as additional inputs to the neural network, yield satisfactory results for identifying tool wear conditions. Lu [63] et al. have used a multi-sensor data fusion intelligent system to detect the state of grinding wheels through mechanical signals and acoustic emission signals. Through the proposed multi-signal processing method based on an artificial immune algorithm, the monitoring accuracy can be continuously improved; Chen Quntao has used sound sensors and vibration sensors as signal detection components to analyze the technical problems related to tool damage monitoring during milling process using multi-sensor information fusion technology. Chen Gang et al. of Beijing Institute of Technology have collected cutting during cutting. The force and vibration signals were extracted by time domain analysis, frequency domain analysis and wavelet packet analysis. Then, the feature selection algorithm in the filtering method was used to complete the feature selection. Finally, the input was input to the three-layer BP neural network for training. The results showed that the recognition accuracy is higher after feature fusion, whether it is a cutting force signal or a vibration signal.

## 4. Reliability Assessment of Machining Equipment based on Condition Information

The research on the reliability of machining equipment can be divided into two types. One is to regard a certain equipment or machine tool as a product, and regard the service life or maintenance characteristics of the equipment as the research content; the other is to regard the machining equipment as the entire manufacturing system, taking the equipment status identification information, process parameter information and the product quality into account, and evaluate the ability of the system to achieve functionality.

### 4.1. Operational Reliability Assessment Method of Machining Equipment

The dynamic signal of the mechanical equipment can effectively reflect the intrinsic state characteristics. It contains important information about the health status of the equipment. The research process of equipment operation reliability evaluation based on state information is as follows: Detecting the dynamic signals of the running mechanical equipment, extracting the characteristic information related to the state change, and establishing the fuzzy function relationship between signal data and equipment reliability through statistical analysis, pattern recognition and machine learning techniques. The reliability evaluation and residual life prediction of the monitored object are realized. Many studies in related fields have demonstrated that reliability analysis methods based on state information have been widely recognized.

The operation reliability assessment based on state information is mainly divided into two different categories. One is the probability statistical techniques, such as the proportional failure rate model, the proportional covariate model, the logistic regression model, the Wiener process, the Gamma process and the Bayesian inference. They are applied to the reliability assessment and residual life prediction of mechanical equipment. The other is the artificial intelligence techniques, such as fuzzy decision trees, Artificial Neural Networks and Support Vector Regression. The evaluation model and the prediction model are trained by historical fault data [64].

### 4.1.1. Probability and Statistics Method

The state information characteristics can reflect the condition of the equipment. So, there must be some mapping relationship between them. In the absence of empirical information to make it difficult to determine the failure threshold, a small amount of historical failure data from similar equipment can be used to establish a mathematical model which describes the potential failure mechanism [65]. There are several representative models, such as the proportional failure rate model, the proportional covariate model and the logistic regression model.

Jardine et al. [66] have proposed a method for analyzing the failure data of aircraft and marine engines using the Weibull proportional failure rate model. The metal element content of the engine was used as a covariate. The research team developed the EXAKT software based on PHM technology. Liao et al. [67] have used the PHM model and the logistic regression model to evaluate the performance degradation and failure probability of the bearing and predict the remaining life of the bearing. Caesarendra et al. [68] have combined the supervised learning method of the correlation vector machine with the logistic regression model to analyze the degradation data of the bearing to predict its failure probability. Chen et al. [69] have evaluated the reliability of the tool and predicted the remaining life of the tool by monitoring the vibration signal during the cutting process and combining the logistic regression model. Combined with Gaussian Mixture Models, Yu [70] has used the negative log-likelihood probability as the degradation evaluation index of rolling bearings and realized the degradation state evaluation of rolling bearings successfully. Zhang et al. [71] have proposed a hybrid Weibull Proportional Hazard Model and realized the degradation state assessment and life prediction of high-pressure water pumps. They also demonstrated that the model is suitable for evaluating the performance degradation trend of mechanical equipment systems with multiple failure modes.

The common idea of the above model is to use the condition monitoring feature index of the historical failure equipment and its corresponding time information to estimate the parameters of the degenerate trajectory model, and then obtain a mapping relationship between the state feature and the health degree. After the new state feature sequence is obtained, the reliability analysis can be performed. Figure 5 shows a flow chart of the reliability evaluation method based on the proportional covariate model. The model established by this method is simple and intuitive. It can quickly determine the degraded trajectory by directly fitting the degraded data. If there is a good understanding of the failure mechanism of the product, the degradation trajectory model can be determined according to the corresponding physico-chemical process by deeply analyzing the failure mechanism of the product. For the mixed effect model of a single random coefficient, the parameter estimation is relatively simple. For the mixed effect model with multiple random coefficients, the calculation process is slightly complicated. The stochastic process model starts with the data. And the degradation model can be given directly according to the characteristics of the data. Then the degradation analyses are performed through data fitting. This method can well describe the uncertainty of the experimental data.

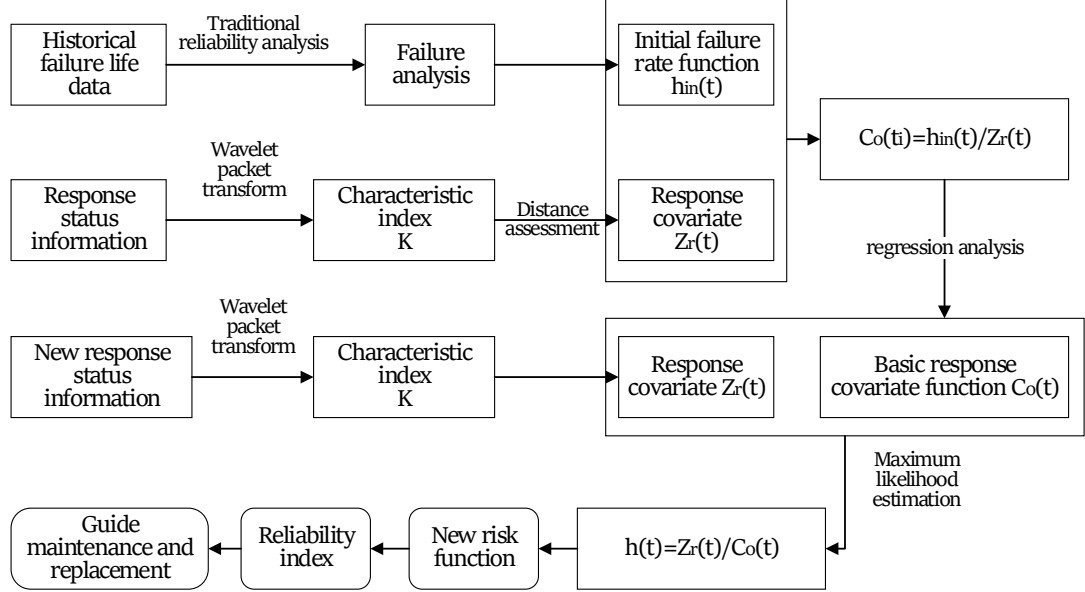

**Figure 5.** Flow chart of reliability evaluation method based on proportional covariate model.

The above methods have been widely used in the field of mechanical product reliability. But these methods are more based on historical data. In a complex operating environment, due to some external factors, it may cause abnormalities in product performance degradation. These methods cannot optimize the model based on the current state, making the model less accurate.

### 4.1.2. Artificial Intelligence Method

Machine learning is a means of converting data into information. The main methods are inductive learning and analytical learning. The "clustering" algorithm can only classify data into different categories. If you want to achieve the purpose of prediction, you need a "classification" algorithm [72–75]. The decision tree is an instance-based inductive learning algorithm [76,77]. Quinlan has proposed the ID3 algorithm and the C4.5 algorithm. In order to adapt to the needs of processing large-scale data sets, several improved algorithms were proposed. Wang Jisheng et al. [78] have used support vector machines to classify the two cutting conditions of normal cutting and tool wear. Hu Lei [79] has proposed an online detection algorithm for turbo pump test data. Based on the optimal algorithm, a complete sample region description model for the time domain statistical characteristics was successfully established. Chang Qi [80] has analyzed the pattern recognition methods of artificial neural network, radial basis function network and least squares support vector machine, and compared the classification effects of those three methods. The process of the operational reliability assessment based on artificial intelligence technology is shown in Figure 6.

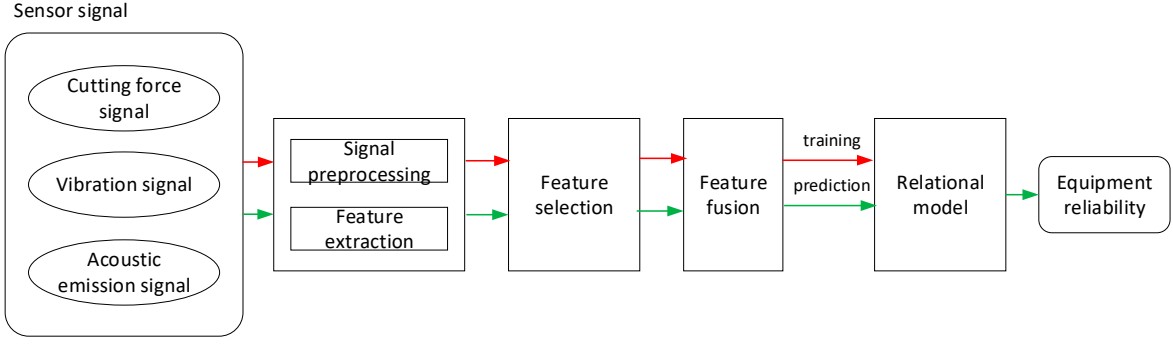

**Figure 6.** Operational reliability assessment based on artificial intelligence technology.

In the early processing of the signal, it mainly consists of two parts. One is to transform the original information of the running state into a set of features with obvious physical meaning or statistical significance, namely signal characteristics. The second is to select a set of the most meaningful feature subsets from the feature set and find the key features, that is, the state feature quantity selection. Feature extraction is a method of transforming a measurement set of patterns to highlight representative features of a pattern, and methods of extracting desired features by image analysis and transformation. Signal processing technology has a long history of research, highlighting the characteristics of signals by transforming the domain in which the signal is located. Commonly used signal feature extraction methods are sometimes domain method, frequency domain method, time-frequency method, and the like. The signal processing techniques used in traditional state monitoring systems focus on statistical analysis in the time domain and frequency domain [81–83]. In recent years, signal analysis technology has developed in the direction of intelligent technology for time-frequency analysis, with time-frequency analysis becoming the mainstream direction of signal analysis [84–101]. The purpose of feature selection is to select a set of most statistically significant subsets of features from the feature set to achieve dimensionality reduction. As can be seen from the literature [102–105], the definitions of feature selection are basically based on the classification accuracy rate and class distribution.

Scholars have applied artificial intelligence technology to the establishment of equipment operation reliability models, and have proposed reliability models based on neural network, support vector machine and hidden Markov. In 1999, Chinnam [106] used the thrust and torque in the processing of high-speed steel drill bit as the operating index, then modeled it using a polynomial regression model with Gaussian white noise. In 2001, Lu et al. [107] used a second-order exponential smoothing time series prediction model for real-time reliability prediction. In 2008, Nagi et al. [108] extracted the vibration signal from a bearing as a degradation indicator of the bearing, established a neural network degradation prediction model, and predicted the life distribution of the bearing. In 2011, Gasperin et al. [109] extracted features from real-time monitored gearbox vibration signals and estimated the remaining service life and its confidence interval using a time series prediction method based on state space model.

Many other scholars have also conducted more research in this regard. Fong [110] pointed out that due to the difference of operating environment and conditions, the degradation situation of each individual equipment is unique. The state monitoring information during the operation can effectively evaluate and predict the reliability of the equipment. Fang Mingjie et al. have proposed a reliability prediction method for hydraulic system of CNC street grinders based on running state information and support vector regression, including the selection of state feature indicators, the calculation method of reliability and the support vector regression modeling method. He Zhengjia et al. [111,112] have proposed using the operating state information to realize the operation reliability evaluation under small sample conditions and expounded the technical route and connotation of mechanical equipment operation reliability based on dynamic modeling, fault mechanism analysis, signal processing and fault feature extraction. Hua et al. [113] have considered the lack of empirical information and many uncertainties in equipment operation when dealing with state monitoring data. A reliability prediction method for adaptive failure threshold was proposed and successfully applied with bearings and high-pressure derusting pumps. Wu Jun et al. [114] have proposed a HMM-based equipment operation reliability prediction method and verified the feasibility of the method by numerical control milling machines. Dong [115] comprehensively discusses the concepts, modeling algorithms and application in the health prediction of HMM and HSMM.

This method is very dependent on data samples. It is a method based on a learning model. Through the defined learning algorithm, the intrinsic relationship between the input features and the output target represented by the data samples is learned. The quality of the data set determines the quality of the relational model. For the data acquired by the cutting process, it often has strong uncertainty and incompleteness due to the interference of complex environment. Therefore, the critical

issue of a process equipment reliability assessment based on state information is how to obtain high quality data samples. At present, there are many researches on the theory and methods of equipment operation reliability based on artificial intelligence technology that is gradually being developed into practical applications.

### 4.2. Mission Reliability Assessment Method of Machining Equipment

In recent years, some scholars have been studying the mission reliability of manufacturing systems. The function of machining equipment is to achieve high productivity, meet the requirements of product quality and meet the delivery time. Their research is not only based on equipment reliability, but from the functional point of view, must consider the ability of machining equipment to meet customer needs or complete the corresponding functions.

Miriyala K et al. has conducted a study on the reliability assessment of flexible manufacturing systems. The article mentions that the reliability of a flexible manufacturing system can be understood as "the steady state probability that the system can perform all given requirements" [116]. The research has certain representativeness. Based on the flexibility of the processing path, the paper establishes a process-spanning graph model and proposes an algorithm based on this model.

Bao Weiwei et al. [117] have defined manufacturing system reliability, that is, the ability of the manufacturing system to complete the scheduled production tasks as required during the planned production cycle. They completed the analysis of the manufacturing system functions. The main function of the manufacturing system is manufacturing capacity (efficiency), manufacturing cycle (delivery) and manufacturing quality. Finally, a framework model for manufacturing system reliability analysis was established, which provides research ideas for reliability research.

Chi Yongjiao et al. [118] have focused on the construction of the mission reliability model based on process information fusion. Firstly, the definition and the role of information fusion are clarified, as shown in Figure 7. Then, the mission reliability of the machining equipment is comprehensively analyzed based on the equipment status identification information, process design analysis and product quality. The reliability model is established by using vibration signal characteristics and tool wear values and the reliability curve is drawn. Finally, the equipment reliability model and processing task information will be determined. Combined with the initial machining condition, the mission reliability assessment of the entire machining process is achieved.

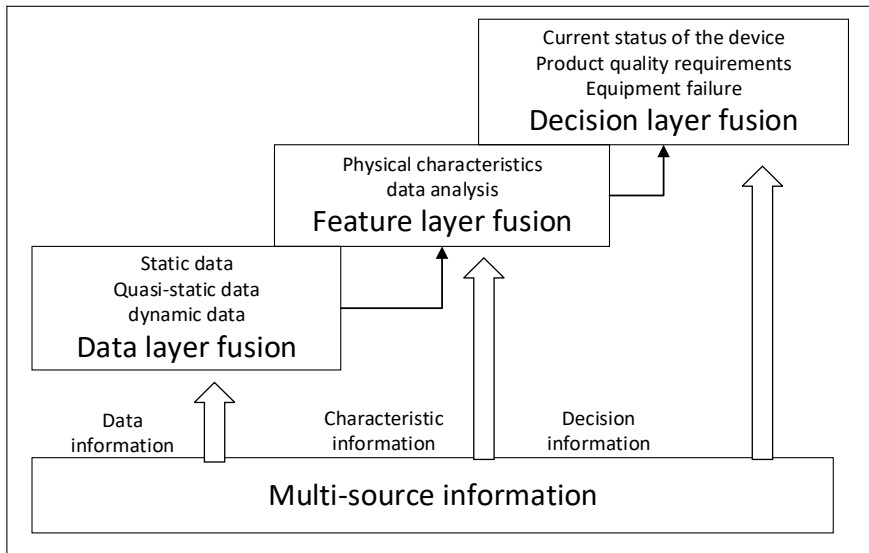

**Figure 7.** Application flow chart of information fusion.

## 5. Summary and Outlook

### 5.1. Outlook on Future Challenges and Trends

The method for equipment reliability assessment using the condition monitoring method has entered the public's field of vision. The dynamic information reflecting equipment health status can be obtained through condition monitoring technology. Considering the characteristics of machining equipment, this method can be further developed. From the current development situation, the tendency of condition monitoring technology and reliability assessment methods of the machining equipment in the future can be summarized as follows.

(1)   Real-time signal acquisition, processing and decision-making methods during machining

At present, the existing research on data acquisition technology of cutting process mainly focuses on the application of wired sensors. Since wired sensors are difficult to apply in the actual data acquisition of the cutting process, it is important to consider the application of wireless sensing technology to improve the practicality of the state monitoring.

Integration and intelligence are important development directions of future condition monitoring systems. It is necessary to make full use of artificial intelligence technology to improve the self-learning, self-determination and self-decision abilities of the monitoring system. In order to improve the accuracy and rapid response abilities of equipment monitoring, new intelligent mode algorithms need to be applied.

(2)   Health monitoring of machining equipment based on full life cycle

The normal operation of the condition monitoring system requires the support of multiple modules, such as sensing, excitation, etc. The later transformation of the equipment not only reduces economics, but also affects the structure and function of the equipment due to space constraints. Therefore, the health monitoring of machining equipment based on the whole life cycle is a way to develop equipment condition monitoring. Monitoring components needs to be integrated at the beginning of the design. In addition, the current research on machining equipment mainly focuses on tool wear and bearing damage. Condition monitoring for other components such as electric motors and tool change systems needs to be further developed.

(3)   Reliability assessment based on multi-source data fusion

At present, the reliability of machining equipment is mainly evaluated by fault data and traditional design reliability models. The use of quality data for the cutting process has just begun. Modern manufacturing systems have a high degree of automation, low failure rates, and few conventional fault data. On the contrary, there are data indicating the quality of the cutting process such as quality inspection and component degradation, as long as the machining equipment is operated. The process quality data can be easily obtained in production. They can reflect the operating condition and reliability of the machining equipment. These valuable data are ignored in the traditional reliability analysis of machining equipment, resulting in many system failures that cannot be diagnosed and predicted in time, and the quality and reliability of products cannot be guaranteed.

Based on the traditional reliability theory, the reliability evaluation of equipment during operation is one of the research hotspots in the field of reliability engineering. Condition monitoring, signal processing, feature index extraction and dynamic modeling are taken as technical routes. The cutting process information, the quality characteristics of the products and the process flow and parameter information are used as multi-source information. They are incorporated into the reliability assessment model and then used for real-time reliability assessment, which is one of the trends in the reliability of machining equipment.

*5.2. Conclusions*

This paper takes the reliability analysis method of mechanical machining equipment based on condition monitoring technology as the main line and carries out literature research from three aspects: Multi-source information in cutting process, failure physical analysis for signal selection and reliability assessment based on condition information.

For the research of multi-source information in the cutting process, this paper reviews the cutting process information, processing product quality characteristics information, processing flow and parameter information of the machining equipment and analyzes the correlation between the three contents in deep. For the failure physical analysis of signal selection, the relationship between the failure mechanism and the output signal is reviewed. For the reliability evaluation of machining equipment based on condition information, two aspects of equipment operation and mission reliability assessment methods was reviewed. It was found that the existing methods are mainly the fusion analysis of cutting process conditions data. It is a feasible and valuable research direction to evaluate the reliability of machining equipment for product quality characteristics, considering the combination of working condition data, product quality information and process information.

**Author Contributions:** J.S. and W.D. conceived the key idea. N.J., Z.L. and J.S. contributed to the writing and editing of the manuscript. W.D., Y.C. and D.X. provided the academic support and checked the manuscript. All authors made contributions to the writing and revising of the manuscript.

**Acknowledgments:** This work was supported by the National Natural Science Foundation of China (No.51705015), the National Science and Technology Major Project (No.2017ZX04008001), and the Fundamental Research Funds for Central University.

**Conflicts of Interest:** The authors declare no conflict of interest.

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
