# Peer review of "Review of Machining Equipment Reliability Analysis Methods based on Condition Monitoring Technology"

_applsci, doi:10.3390/app9142786_

Reviewer 1 Report

The paper topic is important and interesting from a practical point of view. This paper can be published but after the above suggestions:

Review articles must include a Methods section explaining how the literature for review was selected, such as:

* Content analysis;

* Grounded Theory; or

* Discourse Analysis.

 Abstract is reflect the content and summarize the problem, the method, the results, and the conclusions

Paper based on 68 references. This is not enough for review paper. Additionally, I personally feel that this part of paper is not concise enough from a reader’s perspective.

Topic review must provide a comprehensive critical review of recent developments in a specific area or theme that is within the scope of the journal, not only a list of published studies or a bibliometric one. Introduction is expected to have an extensive literature review followed by an in-depth and critical analysis of the state of the art. References section should be extensive about information connecting with cutting process monitoring and research methods. I suggest add information to better describe what other researchers have done in this area. I suggest add important and new articles wrote by prof Szymon Wojciechowski from this topic. In my opinion in paper is lack of metrology and surface quality after cutting. New idea on the world is Information Rich Metrology – in my opinion those information can improve this paper. Please look on papers:

Study on metrological relations between instant tool displacements and surface roughness during precise ball end milling

By: Wojciechowski, S.;

MEASUREMENT   Volume: 129   Pages: 686-694   Published: DEC 2018

Optimisation of machining parameters during ball end milling of hardened steel with various surface inclinations

By: Wojciechowski, S.; Maruda, R. W.; Barrans, S.; et al.

MEASUREMENT   Volume: 111   Pages: 18-28   Published: DEC 2017

The discussion is shallow and needs more details, the observations and future trends. This chapter should be connected with others published papers.

The conclusions are missing;  Please try to emphasize your novelty, put some quantifications, and comment on the limitations based on review.

 Author Response

Reviewer 2 Report

Figure 1- poor image quality. The figure quality must be improved.

Figure 4 - the image and figure name should be on the same page

Figure 5- the text font size and style should be the same with the ones used in the paper text

Upper case letter should not be used for the authors names mentioned in the paper text( for exemple CHINNAM, LU, DONG,.... )

In row 143 the word ”vector” is repeted. Correct the repetition.

Author Response

Round  2

Reviewer 1 Report

Congratulations, the paper is ready for publication